



# **Dynamics of short-term ecosystem carbon fluxes induced by**
# **precipitation events in a semiarid grassland.**
Josué Delgado-Balbuena[1], Henry W. Loescher[2,3], Carlos A. Aguirre-Gutiérrez[1], Teresa
Alfaro-Reyna[1], Luis F. Pineda-Martínez[4], Rodrigo Vargas[5], and José T. Arredondo[6].
[1]Centro Nacional de Investigación Disciplinaria Agricultura Familiar, INIFAP, km 8.5 Carr. Ojuelos – Lagos
de Moreno, 47563, Ojuelos de Jalisco, Jal., México
[2]Battelle, National Ecological Observatory Network (NEON), Boulder, CO USA 80301,
[3]Institute of Alpine and Arctic Research (INSTAAR), University of Colorado, Boulder, CO USA 80301.
[4]Universidad Autónoma de Zacatecas, 108 Calzada Universidad, 98066 Zacatecas, Zacatecas, Mexico
[5]Department of Plant and Soil Sciences, University of Delaware, Newark, DE, USA
[6]Division de Ciencias Ambientales, Instituto Potosino de Investigación Científica y Tecnológica, Camino a la
Presa de San José 2055, Lomas 4ta, 78216 San Luís Potosí, S.L.P., México.
*Correspondence to:* Josué Delgado-Balbuena (delgado.josue@inifap.gob.mx)
**Abstract.** Precipitation (PPT) patterns in semiarid grasslands are characterized by infrequent and small PPT
events; however, plants and soil microorganisms are adapted to use the unpredictable small pulses of water.
Several studies have shown short-term responses of carbon and nitrogen mineralization rates (called the
priming effect or the Birch effect) stimulated by wet-dry cycles; however, dynamics, drivers, and the
contribution of the "priming effect" to the annual C balance is poorly understood. Thus, we analysed six years
of continuous net ecosystem exchange measurements to evaluate the effect of the PPT periodicity, magnitude
of individual PPT events on the daily/annual ecosystem C balance (NEE) in a semiarid grassland. We
included the period between PPT events, a priori daytime NEE rate and a priori soil moisture content as the
main drivers of the priming effect. Ecosystem respiration (ER) responded within few hours following a PPT
event whereas it took five-nine days for gross ecosystem exchange (GEE; such as –NEE = GEE + ER) to
respond. Precipitation events as low as 0.25 mm increased ER, but cumulative PPT > 40 mm that infiltrated
deep into the soil profile stimulated GEE. Overall, ER fluxes following PPT events were related to the change
of soil water content at shallow depth and previous soil conditions (e.g., previous NEE rate, previous soil
water content) and the size of the stimulus (e.g., PPT event size). Carbon effluxes from priming effect
accounted for less than 5% of ecosystem respiration but were significantly high respect to the carbon
balance. In the long-term, changes in PPT regimes to more intense and less frequent PPT events, as expected
by the climate change effect, could convert the semiarid grassland from a slight C sink to a C source.
Keywords: Eddy covariance, net ecosystem exchange, ecosystem respiration, *Bouteloua gracilis*, blue grama,
priming effect, Birch effect.



## 1. **Introduction**

Arid lands comprise a wide range of ecosystem types covering more than 30% of terrestrial land (Lal, 2004). In these ecosystems annual potential evapotranspiration is larger than annual precipitation due to regional atmospheric high-pressure zones (Hadley cells), continental winds, cold oceanic winds and local orographic effects that reduce the precipitation amounts (Maliva and Missimer, 2012). Here, precipitation (PPT) occurs as infrequent, discrete, small (< 5 mm) and unpredictable events (Noy-Meir, 1973; Loik et al, 2004). This results in water-limited ecosystems, where biological activity is restricted to periods of soil water availability (Lauenroth and Sala, 1992). Consequently, the productivity and stability of these ecosystems are more vulnerable to changes in climate, particularly to changes of the historic mean annual PPT (MAP) amounts and the change in the periodicity (frequency) of these PPT events.

Precipitation stimulates short-term changes of carbon and nitrogen mineralization rates because soil microorganisms activate with the increase of soil water content (Turner and Haygarth, 2001). This "priming effect" (Borken and Matzner, 2009) also called the Birch effect (Birch, 1964), describes the soil carbon released from decomposition of heterotrophic sources to the atmosphere following soil rewetting. Amount and timing of PPT events modify the magnitude and duration of the priming effect by modulating soil wet-dry cycles. The size of a PPT event determines the temporal duration and the biotic components that respond to the pulse (Huxman et al., 2004a), and thus, defines the magnitude and direction of $CO_2$ effluxes (Chen et al., 2009). In general, small precipitation events that induce changes in soil humidity at shallow depth do not induce plant activity, but activate soil microorganisms (Collins et al., 2008). On the other hand, successive rewetting cycles reduce carbon mineralization rates as the amount of available organic labile carbon declines (Jarvis et al., 2007). Thus, PPT events after long drought periods (until nine months in semiarid grasslands) trigger larger and longer soil respiration efflux rates compared to consecutive PPT events (Reichmann et al., 2013).

At the ecosystem scale, deserts and grasslands have shown larger $CO_2$ efflux rates after rewetting than temperate ecosystems or croplands (Kim et al., 2012), and in ecosystems with low soil organic carbon content (Bastida et al., 2019). Characteristics and dynamics of these short-term soil C effluxes were addressed by the "Threshold-Delay" model (T-D model, Ogle and Reynolds 2004). The T-D model take the previous environmental conditions, PPT event size, PPT thresholds, and time-delays to inform the time constants that modulate ecosystem responses after a PPT event. Moreover, Huxman et al., (2004a) described the dynamics of the net ecosystem exchange of carbon (NEE) and its components (gross ecosystem exchange = GEE, and ecosystem respiration = ER, such as –NEE = GEE + ER) with parameters of the T-D model (Fig. A1). GEE and ER have different time delays based on threshold PPT quantities and event size, with ER responding to smaller PPT events than GEE (Huxman et al. 2004a). In addition, both GEE and ER have asymptotic responses to large PPT events (the upper PPT thresholds), with an upper ER threshold lower than that found for the GEE threshold (Huxman et al. 2004).

In the semiarid grasslands of Mexico, small PPT events are likely to activate biological soil crusts (BSC) on the soil surface that cover up to 60% of plant interspaces (Concostrina-Zubiri et al., 2014), and to stimulate





ER instead of C uptake.  However, *Bouteloua gracilis* H.B.K. Lag ex Steud (blue grama)*,* the keystone
species in the semiarid grassland of Mexico (Medina-Roldán et al., 2007) may contribute to C uptake because
of its adaptations to take advantage of smaller PPT events (Sala and Lauenroth, 1982, Medina-Roldán et al.,
2013).  Understanding disturbances of ecosystem processes (C fluxes) due to changing regional PPT pattern
in semiarid grasslands is particularly salient given that the global circulation models forecast a 10% to a 30%
reduction of summer and winter precipitation, respectively at the end of the 21$^{st}$ Century (Christensen et al.,
2007), and the PPT patterns is forecasted to have fewer events with more water quantity per event (Easterling
et al., 2000).
Thus, the objective of this study was to evaluate the effect of PPT periodicity and magnitude of individual
PPT events and a priori soil moisture conditions on daily and annual ecosystem C balance (NEE) for the
semiarid grassland in Mexico.  Over a six-year study period, we examined event-based PPT amount, the
period between PPT events, a priori daytime NEE rate and a priori soil water content at two depths as the
main drivers of daily mean NEE change rate.  Because we were interested on short-term NEE change and its
components, only short-term NEE change within few days following a PPT event were evaluated.  Effects on
daily mean GEE (GEE = -NEE + ER) was also evaluated at the beginning of the growing season.  Based on
the T-D model (Ogle and Reynolds, 2004), we expect that; 1) semiarid grassland will exhibit a quick response
(short time-delay) to small PPT events (Low PPT threshold) through positive NEE fluxes (C release, H1).
Moreover, 2) ER and GEE (C release and C uptake, respectively) will differ in their time response and PPT
thresholds, with shorter time-delays and lower PPT thresholds for ER than GEE (H2).  This response is
because of small PPT events should enhance ER mainly through heterotrophic respiration of soil surface
microorganisms that are activated within one hour after wetting (Placella et al., 2012), whereas larger PPT
events are required to reach roots at deeper soil profiles and that plants need longer times for start growing.
On the other hand, we expect that, 3) size and timing of PPT patterns will modulate the magnitude of C
efflux; therefore, large precipitation events after long dry periods will release more $CO_2$ than small or
consecutive PPT events (H3). Finally, we expect, 4) C efflux after PPT events will be a meaningful $CO_2$
source to the atmosphere in the semiarid grassland which will decrease the annual net C uptake of the
ecosystem (H4).
**2. Materials and methods**
**2.1 Site description**
The study site is located on a shortgrass steppe, within the Llanos de Ojuelos subprovince of Jalisco state,
Mexico.  The shortgrass biome in Mexico extends from the North American Midwest along a strip that
follows the Sierra Madre Occidental through the Chihuahuan Desert into the sub-province Llanos de Ojuelos.
Vegetation is dominated by grasses, with *Bouteloua gracilis* (Willd. ex Kunth) Lag. ex Griffithsas the key
grass species, forming near mono-specific stands.  The region has a semiarid climate with mean annual
precipitation of 424 mm ± 11 mm (last 30 years, Delgado-Balbuena et al., 2019) distributed mainly between
June and September and with 6 to 9 months of no rain.  Winter-summer rain accounts for < 20% of the total



annual precipitation (Delgado-Balbuena et al., 2019).  Mean annual temperature is 17.5 ± 0.5 °C.  The
topography is characterized by valleys and gentle rolling hills with soils classified as haplic xerosols
(associated with lithosols and eutric planosols), and haplic phaeozems (associated with lithosols) (Aguado-
Santacruz, 1993).  Soils are shallow with average depth of 0.3-0.4 m containing a cemented layer at ~ 0.5 m
deep, with textures dominated by silty clay and sandy loam soils (Aguado-Santacruz, 1993).
The study site is a fenced area of ~64 ha of semiarid grassland under grazing management.  A 6 m high tower
was placed at the center of the area of interest to support carbon-energy flux measurements and
meteorological instruments as well.  That location allowed an ever-changing and integrated measurement
footprint of 320 m, 410 m, 580 m, and 260 m from the tower according to the N, E, S, and W orientations,
respectively.

## 2.2 Meteorological and soil measurements

Meteorological data was collected continuously at a rate of 1 s and averaged at 30 min intervals using a
datalogger (CR3000, Campbell Scientific Inc., Logan, Utah).  Variables measured included air temperature
and relative humidity (HMP45C, Vaisala, Helsinki, Finland) housed into a radiation shield (R.M. Young
Company Inc., Traverse City, MI), incident and reflected shortwave and longwave solar radiation (NR01,
Hukseflux, Netherlands), and photosynthetic photon flux density (PPFD, PAR lite, Kipp and Zonen, Delft, the
Netherlands).  Soil variables were measured at a 5 min frequency and averaged at 30 min intervals.  These
included volumetric soil water content (CS616, Campbell Sci., Logan, UT) positioned horizontally to 2.5 cm
and 15 cm deep, average soil temperature of the top 8 cm soil profile, and soil temperature at 5 cm deep
(T108 temperature probes, Campbell Scientific Inc., Logan, UT).  Soil temperature variables were acquired
with another datalogger (CR510, Campbell Scientific Inc., Logan, UT). Precipitation was measured with a
bucket rain gauge installed 5 m away from the tower (FTS, Victoria, British Columbia, Canada) at 1 m.a.g.l.

## 2.3 Net ecosystem $CO_2$ exchange measurements

An open path eddy covariance system was placed at 3 m high to cover a fetch of 300 m and used to measure
NEE over the semiarid grassland.  The system consisted of a three-dimensional sonic anemometer (CSAT-3D,
Campbell Sci., Logan, UT) for measuring wind velocity on each polar coordinate ($u,\ v,\ w$) and sonic
temperature ($\theta s$), and an open-path infrared gas analyzer (IRGA, Li-7500, LI-COR Inc., Lincoln, NE) to
measure $CO_2$ and water vapor concentrations.  Instruments were mounted in a tower at 3 m above the soil
surface oriented towards the prevailing winds. The IRGA sensor was mounted next to—and 10 cm offset from
the anemometer transducers, the center of the IRGA optical path was centered with the distance between the
vertically oriented sonic transducers and tilted 45° to avoid dust and water accumulation in the IRGA optical
path.  Digital signal of both sensors was recorded at a sampling rate of 10 Hz in a datalogger (CR3000,
Campbell Scientific Inc., Logan, UT) (Ocheltree and Loescher 2007).  NEE was estimated as:
$$NEE = \overline{w'CO_2'} \tag{1}$$





overbar denotes time averaging and primes are the deviations of instantaneous values (at 10 Hz) from a block-
averaged mean (30 min) of vertical windspeed (w, m s$^{-1}$) and molar volume of $CO_2$ (µmol $CO_2$ m$^{-3}$),
respectively. Micrometeorological convention was used, where negative NEE values stand for ecosystem C
uptake (Loescher et al., 2006). We did not estimate a storage flux because of the low vegetation stature and
well mixed conditions; therefore, we assumed it would be 0 over a 24-h period (Loescher et al., 2006).
**2.4 Data processing**
Raw eddy covariance data were processed in EdiRe (v1.5.0.10, University of Edinburgh, Edinburgh UK).
Wind velocities, sonic temperature, [$CO_2$], and [$H_2O$] signals were despiked, considering outliers those values
with a deviation larger than ±8 standard deviations. A 2-D coordinate rotation was applied to sonic
anemometer wind velocities to obtain turbulence statistics perpendicular to the local streamline. Lags
between horizontal wind velocity and scalars were removed with a cross-correlation procedure to maximize
the covariance among signals. Carbon and water vapor fluxes were estimated as molar fluxes (mol m$^{-2}$ s$^{-1}$) at
30 min block averages, and then they were corrected for air density fluctuations (WPL correction, Webb et al.
1980). Frequency response correction was done after Massman (2000). Sensible heat flux was estimated
from the covariance between fluctuations of horizontal wind velocity (w') and sonic temperature (θ'$_s$). This
buoyancy flux was corrected for humidity effects (Schotanus et al. 1983, Foken et al., 2012).
Fluxes were submitted to quality control procedures, i) stationarity (<50%), ii) integral turbulence
characteristics (<50%), iii) flags of IRGA and sonic anemometer (AGC value<75, Max CSAT diagnostic flag
= 63) which are strongly related with advices of problem measurement due to rain events, iv) screening of
flux values into a logical magnitudes (±20 µmol $CO_2$ m$^{-2}$ s$^{-1}$), and v) a threshold u*= 0.1 m s$^{-1}$ was used to
filter nighttime NEE under poorly developed turbulence. This threshold was defined through the 99%
threshold criterion after Reichstein et al. (2005).
Temporally integrated estimates are noted throughout this paper. Because of GEE cannot be measured
directly, it was estimated by ER withdrawal from -NEE. The ER was estimated in two ways, 1) it was
estimated from light-response curves (see below), and 2) it was determined from nighttime NEE data (under
PPFD < 10 µmol m$^{-2}$ s$^{-1}$ light conditions). Different ER estimation method is indicated throughout the paper.
For identifying changes induced by PPT events on GEE and ER, daytime and nighttime NEE data on a one
day-window was adjusted with a rectangular hyperbolic response function to photosynthetic photon flux
density (PPFD; Ruimy et al. 1995).
$$NEE = \frac{\alpha * PPFD * A_{max}}{\alpha * PPFD + A_{max}} + ER \qquad\qquad (2)$$
where, α is the apparent quantum yield (µmol $CO_2$ m$^{-2}$ s$^{-1}$/ µmol $CO_2$ m$^{-2}$ s$^{-1}$), $A_{max}$ is maximum
photosynthetic capacity (µmol $CO_2$ m$^{-2}$ s$^{-1}$), and *ER* is the ecosystem respiration (µmol $CO_2$ m$^{-2}$ s$^{-1}$). Due to
Amax is calculated to unrealistic "infinite" PPFD, we calculated a more realistic maximum photosynthetic
capacity ($A_{2500}$), which is maximum photosynthesis at 2500 µmol m$^{-2}$ s$^{-1}$. Changes and transitions from ER
dominated NEE fluxes to C-gain processes (GEE) were verified with the shape of the light response curve.



**2.5 Gap filling procedures and characterization of PPT events**
Data gaps shorter than two hours were linearly interpolated, whereas gaps larger than two hours were left as
empty data. Only daytime-NEE data were used for most of the analysis because of nighttime NEE is
subjected to quality problems, which include poor developed turbulences caused few 30-min periods with
available data and showed strong divergence from NEE averages if the whole night cycle is not similarly
represented among days. Daily mean ER derived from nighttime NEE data were used for analysis when more
than 50% of the data was available after QA/QC procedures. The NEE related PPT events were selected for
analysis based on data quality and availability to evenly cover the daytime cycle (on average more than 85%
of NEE data) and then averaged through the day. The daytime-scale was selected to avoid confounding
diurnal NEE variability and to achieve robust analyses. All precipitation events between 2011 and 2016 were
isolated and then filtered by the number of half-hours accounted for mean daily fluxes.
The C flux one day before the PPT event was taken as the reference C flux. Event-response effect ("priming
NEE effect") was measured as the difference between mean daytime NEE post-event and mean daytime NEE
pre-event, such that.
$\Delta NEE = NEE_{post-event} - NEE_{pre-event}$ (3)
where, NEE is the daytime NEE average ($\mu mol\ m^{-2}\ s^{-1}$).
The same method was used to calculate changes of soil water content at 2.5 and 15 cm depth ($\Delta VWC_{2.5}$ and
$\Delta VWC_{15}$, respectively), and change of photosynthetic photon flux density ($\Delta PPFD$)
Intervals between PPT events (hereafter inter-event periods, IEP) were counted in days from the last PPT
event, regardless of its magnitude.
Enhanced vegetation index (EVI) of 250 m spatial resolution and 8 day time-resolution from NASA's MODIS
instruments was used as an approximation of plant leaf activity. The Savitzky-Golay (Yang et al., 2014) filter
was used to eliminate outliers of EVI derived from adverse atmospheric conditions.
According to the model, where previous conditions are determinant of carbon fluxes, data were divided in
"fluxes dominated by photosynthesis (carbon uptake)" and "fluxes dominated by ecosystem respiration
(carbon efflux)". A threshold of -1 $\mu mol\ m^{-2}\ s^{-1}$ of average previous daytime $CO_2$ flux was used to divide data.
This was done to avoid confounding factors, because of environmental drivers of photosynthesis and
respiration may differ in magnitude and direction. Moreover, under photosynthetic conditions is hard to
identify if a positive change of NEE (less photosynthesis) was due to an increase of soil respiration or a
dampening of photosynthesis by less available radiation under cloudy conditions.
**2.6 Statistical analysis**
Boosted regression trees analysis (BRT; Elith and Leathwick, 2017) were developed to identify the most
important variable controlling the priming C effect and thresholds of this response. BRT analysis also were
used to identify the form of function, i.e., whether relationship between independent variables and the priming
effect was linear, exponential, sigmoidal, peak from, etc. Independent variables included PPT event size, inter
event-periods (IEP), a priori, current, and change of volumetric water content (VWC) at two depths (2.5 and





15 cm), soil temperature, previous daytime NEE, enhanced vegetation index (EVI) and change in
photosynthetic photon flux density (ΔPPFD).  For BRT analysis, data was divided in "photosynthesis
dominated" and "respiration dominated" data. On the other hand, for identify delays between C fluxes
(ecosystem respiration and gross primary productivity) and precipitation events, a cross correlation analysis
was done. For cross correlation, parameter of the light response curve was used; the ER was used to identify
delays between ecosystem respiration and soil water content at 2.5 cm, and $A_{2500}$ was used to identify delays
between gross ecosystem productivity and soil water content at 15 cm, because of ER and $A_{2500}$ were better
correlated with soil volumetric water content at 2.5 and 15 cm, respectively. All these variables were
detrended before cross-correlation analysis. Finally, linear correlation analyses were performed among
environmental variables and priming effect and nighttime ER, and among independent variables to test for
autocorrelations. The "gbm" package (The R core team) was used for performing BRT analysis, whereas the
"astsa" package for R was used to conduct cross correlation analyses.
**3. Results**
**3.1 Precipitation pattern**
Cumulative precipitation for 2011 (288.5 mm) was below the 30-y average for the site (420 mm) and was the
worst drought of the last 70-y.  In contrast, 2012 received less PPT (393.2 mm), and 2014 and 2016 received
more PPT (528.5 and 436 mm, respectively) than average, whereas 2013 (601.6 mm) and 2015 (785.9 mm)
were very humid years (Fig. 1).  The 6-y differed in precipitation frequency, but they were similar in the size
of PPT events with ~60% of the PPT events < 5 mm (Fig. 2a).  However, notwithstanding the lower
proportion of larger size PPT events (PPT events > 5 mm), they summed similar or even more amount of
water than small PPT events (Fig. 2b).  Overall, precipitation pattern was characterized by short inter event
periods with 60% of PPT events falling consecutively (IEP <5 days; Fig. 2c).
Soil saturated after large or recurrent PPT events. Largely, soil moisture was maintained over a 10% in the
wettest years, with the largest peak reaching a 40% in summer 2014 (Fig. 1b).  Most VWC variability was
observed at 2.5 cm rather than 15 cm depth and it was better correlated with precipitation amount per event (p
< 0.05, $R^2$ = 0.72, Fig. 2d) increasing 0.3 % per mm of precipitation.  PPT events of 0.25 mm increased the
$VWC_{2.5}$ in ~1-2%, but this increase lasted for less than one hour, whereas $VWC_{15}$ increased after PPT ~5 mm
(data not shown). Additionally, PPT events and soil moisture dynamics at 15 cm depth were out of phase (up
to five days between the PPT event and the SWC15 peak, Fig. 2e)
A total of 256 events from this 6-y study were used for statistical analysis. A sample of 100 PPT events was
used for the respiration dominated fluxes (>-1.0 μmol m$^{-2}$ s$^{-1}$), and 156 PPT events for the photosynthesis
dominated fluxes (>-1.0 μmol m$^{-2}$ s$^{-1}$). Small precipitation events dominated in our database but represented
well the precipitation pattern of the site.  The sample was integrated by events in the range from 0.25 to 57.1
mm, and a mean of 5.7 ± 0.53 mm (mean ± 1 SE). Large PPT events occurred after short inter-event periods,
and small PPT events were preceded by long inter-event periods.  Medium PPT events after long inter-event
periods were rare, and extreme large PPT events after long inter-event periods were not observed (Fig. 2f).



The size of the precipitation event (PPT) and previous soil water content at 2.5 cm depth (preVWC$_{2.5}$)
explained a large variation of change in soil water content at 2.5 cm depth ($\Delta$VWC$_{2.5}$; R$^2$ = 0.54; Fig. 2d).
Best correlation among variables was observed between previous soil water content and soil water content at
different depths; for instance, VWC$_{15}$ and pre VWC$_{15}$ (R$^2$ = 0.84), between the same variables but at 2.5 cm
(R$^2$ = 0.81). The change in NEE (priming effect) has not a strong relationship with any single variable (Fig.
A2).

**3.2 Time delays and thresholds.**

The minimum PPT event that altered NEE rates was 0.25 mm. Overall, the analysis of half hour fluxes
showed almost instantaneous positive response of NEE to PPT event that exponentially decreased over time
into a half to two hours after the PPT event (Fig. A3). ER rates increased after 0.25 mm PPT events, but we
detected a different threshold for GEE where either a larger PPT event or multiple consecutive events (*e.g.,* >
40 mm, Fig. 2a) was needed, and showed a delay of ~5 days after the positive change in VWC at the 15 cm
depth, this at the beginning of the growing season (Fig. 3a, b).
Cross-correlation analysis of light-response curve parameters showed no lags between ecosystem respiration
(ER) and volumetric soil water content at 2.5 cm. (Fig. 3a), whereas there was a lag of 9 days between
photosynthetic capacity at 2500 PPFD (A$_{2500}$; Fig. 3b) and soil water content, which was larger than the
observed at several precipitation evets of 2013 (Fig. 2a,b).
The BRT analysis showed sigmoidal relationships between the priming effect and environmental variables
with different thresholds. At the respiration-dominated period, a minimum change of soil volumetric water
content at 2.5 cm affected positively the carbon flux, but a change larger than 8% in this variable did not
induce a larger C efflux (upper threshold; Fig. 4). On the other hand, C priming effect was larger under
neutral previous NEE (preNEE~0) and decreased in magnitude as preNEE becomes more positive (Fig. 5).
Moreover, previous dry conditions at shallow soil depth promoted larger C efflux by the priming effect, and
this effect decreased as soil previous conditions were wetter, with a threshold at 15% (Fig. 5). Similar to the
change in soil water content at 2.5 cm, even the lowest PPT event (0.25 mm) caused an increase of C efflux,
but with a threshold between 10 - 15 mm. Precipitation events larger than 15 mm did not enhanced the
priming effect (Fig. 5). In contrast, in the photosynthesis dominated period, larger priming effect was
observed at more negative preNEE (-7 µmol m$^{-2}$ s$^{-1}$) and had no more effect at ~ -4 µmol m$^{-2}$ s$^{-1}$. The priming
effect was enhanced by dry soil conditions at 15 cm depth (< 30%) with a rapid suppression after that. On the
other hand, the priming effect was gradually decreasing with reductions of PPFD.
Nighttime NEE (ecosystem respiration) showed correlation with soil water content at the two depths and EVI;
however, the relationship was linear at low soil water content, reached a maximum at medium values of VWC
and then decreased with minimum values at high soil water content. The largest ecosystem respiration was
observed at higher EVI values (Fig. A4)



**3.3 Dynamics and drivers of the "Priming effect"**
The priming effect lasted longer with initial larger change of NEE, i.e., whereas higher was the priming effect
($\Delta$NEE), the C fluxes lasted more time in returning to initial values (previous to PPT event); however,
decreasing NEE rates were better explained by PPT event size than the initial change of NEE (insert Fig. 4).
For instance, after a 13.7 mm PPT event and initial daytime NEE = 5.1 μmol m$^{-2}$ s$^{-1}$, the C flux exponentially
decreased at a rate of ~50% of its earlier value, whereas with an initial NEE efflux ~2.5 μmol m$^{-2}$ s$^{-1}$, the C
flux decreased at a rate of 100% (Fig. 4). Thus, total C efflux was a contribution of the initial change of NEE
and the time taken to return to basal values (i.e., decreasing rates).
According to BRT analysis, the factor that most influenced the priming effect in the respiration-dominated
period was the change of soil water content at 2.5 cm depth ($\Delta$VWC$_{2.5}$; relative importance, RI = 18%), which
was followed by the a priori NEE (preNEE; RI = 14%), the previous VWC at 2.5 cm depth (RI=14%) and the
size of PPT event (RI = 13%). All the other factors had individual RI values lower than 10% (Table 1; Fig. 6).
Maximum $\Delta$NEE values were observed at i) larger changes of soil water content at 2.5 cm depth (Fig. 6a), ii)
previous neutral NEE (i.e., NEE ~ 0 μmol m$^{-2}$ s$^{-1}$; Fig. 6b), iii) previous dry soil water content at 2.5 cm depth
(Fig. 6c), and iv) with large PPT events (>15 mm d$^{-1}$; Fig. 6d). The priming NEE effect decreased farther than
these limits. In contrast, in the photosynthesis-dominated period, the previous NEE was the most important
factor explaining the "priming effect" (RI=33%), whereas the volumetric water content at 15 cm depth, the
change of photosynthetic photon flux density and the volumetric water content at 2.5 cm depth followed in
importance (Table 1). Larger changes in NEE (priming effect) were observed at i) more negative previous
NEE (i.e., under more photosynthetic activity; Fig. 6e), ii) under drier soil water conditions at 15 cm depth
(Fig. 6f), iii) with larger changes of PPFD (decrease of PPFD; Fig. 6g)), and iv) under air temperature lower
than 16 °C and higher than 19 °C (Fig. 6h). There was a large interaction between preVWC2.5 and PPT for
the respiration-dominated period and between preNEE and $\Delta$PPF for the photosynthesis-dominated period.
**3.4 Contribution of priming effect to carbon balance**
The carbon balance for this six-year period for this site was of -126 g C m$^{-2}$, with 2955 and -3080 g m$^{-2}$ of
ecosystem respiration and gross ecosystem exchange, respectively, and varied from a sink of -107 g C m$^{-2}$ y$^{-1}$
to a source of 114 g C m$^{-2}$ y$^{-1}$ (Delgado-Balbuena et al., 2019). Roughly calculation of carbon efflux due to
priming effect indicated that extrapolation of mean $\Delta$NEE per event and by year, contributes with 142 g m$^{-2}$
for the full six-year period which corresponds to 5% of total ER flux. In this calculation, parameters like
decaying rates, size of PPT event, and previous soil and flux conditions were not considered (modeled) and
was subjected to the number of PPT events. Logically, humid years with a greater number of PPT events have
more contribution of C efflux by priming effect. Each year contributed with less than 30 g m$^{-2}$ y$^{-1}$.



## 4. Discussion

### 4.1 Dynamics of the "Priming effect"

In agreement with the T-D model, NEE exponentially decreased after the PPT pulse (Fig. 5) to almost the pre-PPT NEE rate. The largest C efflux pulses slowly returned to basal C efflux rates and showed larger NEE remnants than the smaller pulses (Fig. 5). This suggests that more persistent VWC quantities achieved with larger size PPT events promoted larger and longer lasting C effluxes. If the event was large enough to maintain VWC above a threshold (e.g., above the wilting point for plants) for a long time, NEE is expected to remain higher than pre-event rates until nutrients or labile C are depleted (Jarvis et al., 2007; Xu et al., 2004). In contrast, when the PPT event is small and the soil remains wet for a short-time, the C flux peak will be small and less persistent because of soil dry-out and the activity of microorganisms it is likely to end before soil nutrients are depleted. Thus, 'priming effect' decaying rates ($-k$) likely are more an issue of water availability than nutrient or C source depletion.

### 4.2 Thresholds and time delays of the "Priming carbon flux effect"

In our study, the NEE increased immediately (short-time delay) after a PPT event, in accordance with (H1). Moreover, the minimum size of an PPT event needed to detect NEE change was as low as 0.25 mm d$^{-1}$, in agreement with (H2). We interpret that immediate daytime PPT induced responses in NEE and ER rates were dominated by heterotrophic respiration and assume that these microbial communities have evolved to take advantage of this short-term water availability. Short-term responses of < 30-min have also been reported in studies that analyzed soil microorganism activity through molecular and stable isotope techniques (Placella et al., 20012; Unger et al., 2010). Fungi and bacteria on the soil surface have the capability for water-induced re-activation within 1 to 72-h after a PPT event (Placella et al., 2012). Immediate positive NEE increase observed in our study (Fig. A.3) may have resulted from such rapid activation of bacteria displaying highest activity 1-h after wetting. Biological soil crust (BSC) are assemblages of microorganism forming crusts on the soil and rock surfaces (Belnap, 2003) common in arid lands. At our site, the BSC covers up to 70% of plant interspaces in grazing-excluded conditions and up to 30% in overgrazed sites (Concostrina-Zubiri et al., 2014) with dominance of actinobacterias (e.g., actinomycetes) and cyanobacterias, which are identified as rapid responders (Bowling et al., 2011). Moreover, Medina-Roldán et al. (2013) at the same study site showed an increase of 36% and 34% of extractable NH4+ and NO3-, respectively, after a PPT event of 10 mm.

The maximum priming NEE effect was identified under changes larger than 8% of soil water content at 2.5 cm, previous dry soil, neutral previous NEE and PPT events > 15 mm. These limits may be defined by several conditions, including; 1) the largest and most intense events did not completely infiltrate into the soil, forming abundant runoff, and moderating the amount of water penetrating the soil profile at similar depth as that found from large-size PPT events, 2) oxygen and CO$_2$ diffusion limitation under high soil VWC dampened soil respiration, 3) all soil aggregates are disrupted at medium soil VWC likely providing no additional nutrient or C substrate at higher VWCs (Bailey et al., 2019; Lado-Monserrat et al., 2014; Homyak et al., 2018; Chen et al., 2019), and 4) a combination of any of these three. Linear relationship between PPT





event size, preVWC$_{2.5}$ and ΔVWC2.5 (Fig. 2d) showed that there was not a strong limitation of water
infiltration into the soil at shallow depths, discarding in some way the first condition, whereas the reduction of
ER rates in nighttime NEE data after VWC2.5> 12%, and daytime ΔNEE reductions under higher
preVWC2.5 supports the second mechanism (Fig. 6, and A4).

**4.2 The ER and GEE threshold and time delays difference**

The smallest PPT events only stimulated ER rates, with no apparent change observed in GEE (Fig. 3). Even a
large PPT event of 20 mm d$^{-1}$ recorded in May 2013 (Fig. 3) did not induce an increase in GEE. In contrast,
larger or consecutive PPT events that reached deeper soil profiles stimulated GEE (cumulative PPT > 40mm).
These results also explain why the a priori soil moisture and the change of VWC (2.5 cm depth) better
explained ΔNEE at the respiration-dominated period, rather than soil moisture at 15 cm depth (Fig 5); this
confirms our notion that soil microorganism activity was the source of the immediate $CO_2$ efflux. In contrast,
VWC at 15 cm depth was the second most important factor explaining priming NEE effect in the
photosynthesis-dominated period. Additionally, the change of PPFD during the photosynthesis-dominated
period affected positively the priming effect (Fig. 6), it means that reduction of carbon uptake by cloudy
conditions was larger than the stimulus of ecosystem respiration by the increase of soil moisture.
The low PPT threshold that stimulated ER agrees with results from other studies in arid ecosystems (and are
even lower). PPT events as small as 3 mm induced respiration of biological soil crusts (Kurc and Small,
2007), and PPT events <10 mm d$^{-1}$ on a shortgrass steppe promoted net loss of C (Parton et al., 2012).
However, the dominant species at our site, *B. gracilis*, was reported to respond to PPT events as small as 5
mm (Sala and Lauenroth, 1982), which was the PPT threshold we were expecting. Instead, this study found
that large or consecutive PPT events had to occur before an effect on GEE was observed (Fig 3).
Nevertheless, it is interesting to note that small PPT events in arid ecosystems that do not lead to C uptake
may alleviate stress after severe droughts, rehydrating plant tissues and helping plants to respond faster after
larger PPT events (Sala and Lauenroth, 1982; Aguirre-Gutiérrez et al., 2019).
Causes of larger time-delays in GEE than ER is likely due to the delay between the PPT event and the
infiltration of water to a given soil layer (e.g., 15 cm depth; Fig. 2e), and the time spent for regrowing of new
roots and leaves (Ogle and Reynolds, 2004). These processes promote C losses rather than C uptake in the
early growing season (Huxman et al., 2004; Delgado-Balbuena et al., 2019). In contrast, ER was primarily
controlled by soil moisture at shallow soil layers that moist immediately after any PPT event and may activate
soil microorganism just few hours after soil wetting as discussed above.

**4.3 Influence of event size and a priori conditions**

The magnitude of the priming effect was determined by the size of the PPT event and mainly by the ΔVWC
as well as the prior condition of the ecosystem (i.e., previous C flux, and previous soil VWC). These results
agree with (H3) that proposed the PPT event size and previous conditions of the semiarid grassland would
control the magnitude of the "priming NEE effect". The a priori VWC offers insight into the potential dry-



wet shock experienced by soil aggregates and microorganism (Haynes and Swift, 1990) and thus accounts for
nutrient and labile C accumulation in soil (Bailey et al., 2019).
Results indicated that larger C effluxes were induced from medium amount of PPT when the previous soil
conditions were dry and had a preceding value of NEE = ~0. Several mechanisms can explain this result: i)
the accumulation of nutrients and labile C into the soil (Schimel and Bennet, 2004) because low activity of
microorganisms (NEE ~ 0) under dry soil (Homyak et al., 2018), ii) if soil VWC is maintained for a long
period above a threshold, then soil microbial activity exhaust labile C sources (Jarvis et al., 2007; Fierer and
Schimel, 2002). Consequently, recalcitrant C sources subjected to microbial decomposition decrease
mineralization rates (Van Gestel et al., 1993).
**4.4 Importance of the priming effect in the annual C balance.**
We expected a significant contribution of C release from the "priming effect" to decrease the net annual C
uptake of the semiarid grassland (H4). Contribution of this short-term C efflux events to annual C balances
accounted for a considerable amount, but it was a small contribution if it is considered into the ecosystem
respiration flux, which was almost 3000 g m$^{-2}$ s$^{-1}$. Notwithstanding its contribution is apparently low (~5% of
ecosystem respiration), it is important considering that the annual C balance (NEE) is a small fraction of the
difference between ER and GEE, thus, a 5% of C released represents up to 500% of the net C uptake during
an almost neutral year and may turn a C sink ecosystem into a net C source. Therefore, we cannot reject H4.
**4.5 Priming effect and climate change perspectives.**
The low $\Delta SWC_{2.5}$ and PPT threshold for respiration suggests that almost all PPT events occurring in the
semiarid grasslands will produce C efflux but will be limited by the characteristics of the PPT pattern and
previous soil conditions at the site. Therefore, we expect that small PPT events with dry previous conditions
or long inter-event periods will limit the priming effect by maintaining the system below threshold conditions.
Moreover, consecutive PPT events or large PPT events should keep soil water content above a threshold that
will promote C uptake by photosynthesis, which in the long term will overcome C loses from the priming
effect. However, climate change scenarios forecast for the semiarid grassland in Mexico a decrease of winter
PPT and the increase of storms with larger inter-event periods, which are conditions for increasing the amount
of C released by the priming effect (Arca et al., 2021; Darenova et al., 2019).
It is necessary a further analysis of the effect of these PPT events on vegetation since productivity will also
depend on PPT event size and will be modulated by previous soil conditions. Additionally, it is likely that
productivity will benefit more on accumulated PPT than respiration. Still, more analysis of projected PPT
scenarios is required to forecast accurately the PPT pattern under more frequent droughts, and to know if the
current PPT pattern of dry-wet years will prevail. In this sense, parameterizing a model like de T-D model
will provide valuable information of more accurate C effluxes from the priming effect and how it will be
affected by changes of precipitation pattern. Only after that, we will be able to predict the course of the
semiarid grassland as a source or sink of C under PPT pattern changes.



## 5. Conclusions

Previous soil water conditions and previous NEE were the most important factors controlling the priming effect in the semiarid grassland. The size of precipitation had an important role in explaining the priming effect but only in the respiration-dominated period. Delays between responses of change at deeper soil layer and for regrowing processes could hide relationship between precipitation and priming effect during the photosynthesis-dominated period. Importance of the priming effect in the carbon balance could be more important under forecasted changes in precipitation pattern by increasing in both frequency and intensity the dry-wet soil cycles. A further analysis of the effect of this change of precipitation patter on ecosystem productivity is necessary before we can conclude about changes in the carbon balance of the semiarid grassland.

*Author contributions*. The study was conceived by JD, TA, HL and RV. JD, TA and CAA get and processed eddy covariance data. JD, TAR, LFM implemented the method and performed the data analyses. TAR and CAA get and processed the Enhanced Vegetation Index data. TA, HL, LFM and RV helped to interpret the results. JD, TA, HL, and RV prepared the first draft, and all authors contributed to discussion of results and the revisions of the paper.

*Competing interests*. The authors declare that they have no conflict of interest.

*Availability* of *data.* The datasets used and/or analyzed during the current study are available from Zenodo *https://doi.org/10.5281/zenodo.7379206*

## Acknowledgments

Authors thank INIFAP for the facilities at CENID Agricultura Familiar research site in Ojuelos, Jalisco, to carry out this study. This research was funded by SEMARNAT-CONACYT, project reference number 108000, CB 2008-01 102855, CB 2013 220788 given to TA, and CONACYT CF 320641 given to JDB. HWL acknowledges the National Science Foundation (NSF) for on-going support under cooperative support agreement (EF-1029808) to Battelle. Any opinions, findings, and conclusions or recommendations expressed in this material are those of the authors and do not necessarily reflect the views of our sponsoring agencies.

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





**Table 1. Relative importance (RI) of the first four most important environmental factors for the "priming CO$_2$**
**effect".**

| Respiration-dominated period | RI |
| --- | --- |
| $\Delta VWC_{2.5}$ | 18.66 |
| preNEE | 14.67 |
| preVWC$_{2.5}$ | 14.08 |
| PPT | 13.64 |
| preVWC$_{15}$ | 8.09 |
| VWC$_{2.5}$ | 7.46 |
| Photosynthesis-dominated period | |
| preNEE | 33.32 |
| VWC$_{15}$ | 12.25 |
| $\Delta PPFD$ | 11.52 |
| VWC$_{2.5}$ | 9.16 |
| Tair | 8.32 |
| preVWC$_{2.5}$ | 7.79 |






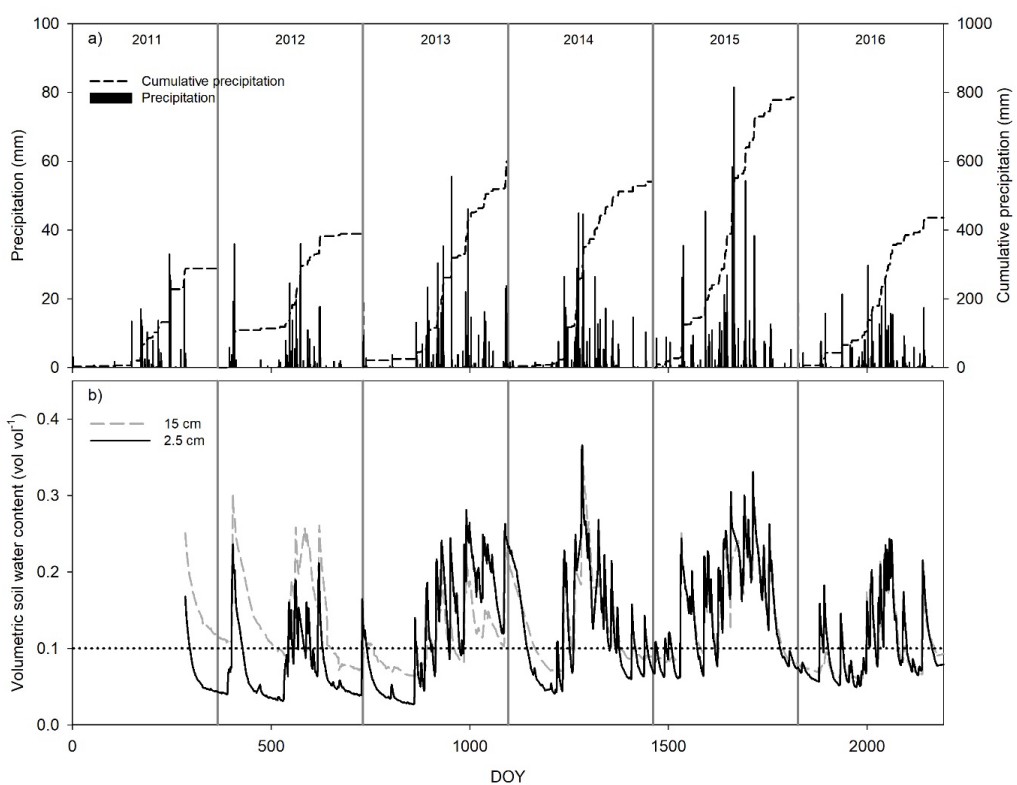


**Figure 1. Seasonal and interannual variation of daily precipitation and cumulative precipitation (a), and volumetric soil water content at 2.5 (black line) and 15 cm depth (gray line; b). Dotted line at 10% of soil water content was depicted as reference.**






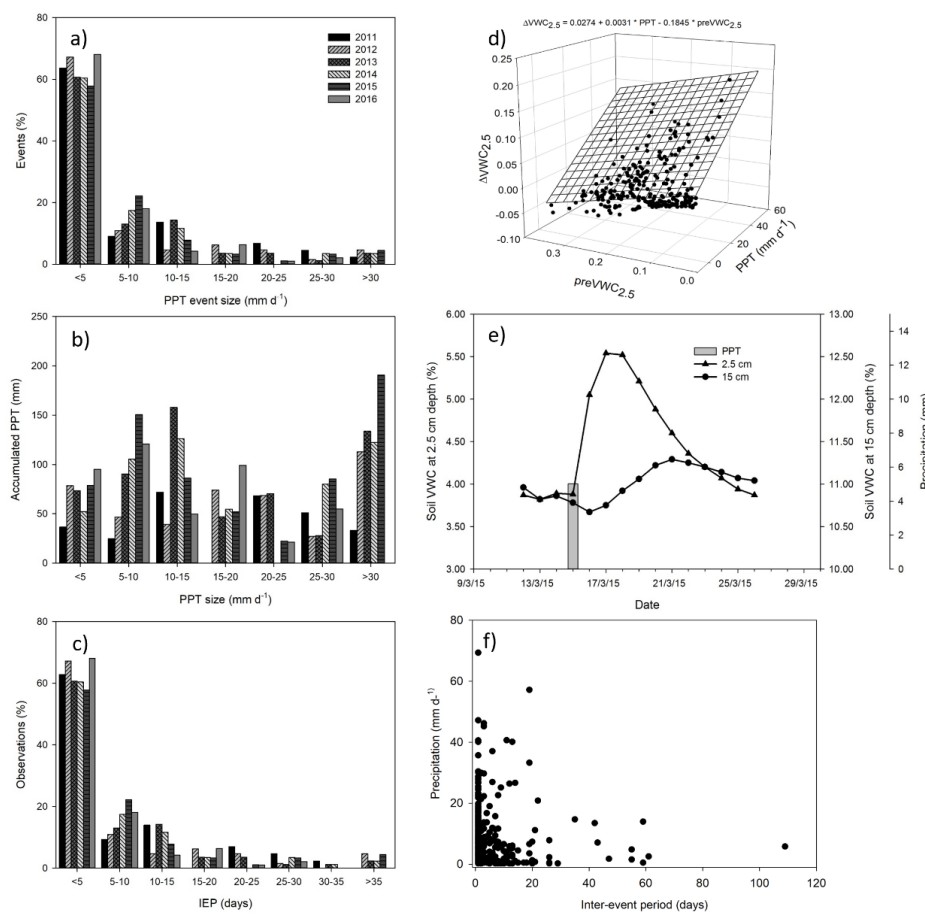


**Figure 2. Characterization of precipitation pattern. Histogram of the size of precipitation events through six years (a), the accumulated precipitation by size of precipitation event (b), and the number (%) of precipitation events by inter-event period classes (IEP, days; c). Relationship between size of precipitation event (mm d⁻¹), previous volumetric soil water content at 2.5 cm depth (v/v) and the change in soil volumetric water content at 2.5 cm depth (v/v). Dynamic of soil water content at two depths (2.5 and 15 cm) after a precipitation event of 5 mm through the time (e), and relationship between inter-event period and the size of precipitation event (f).**



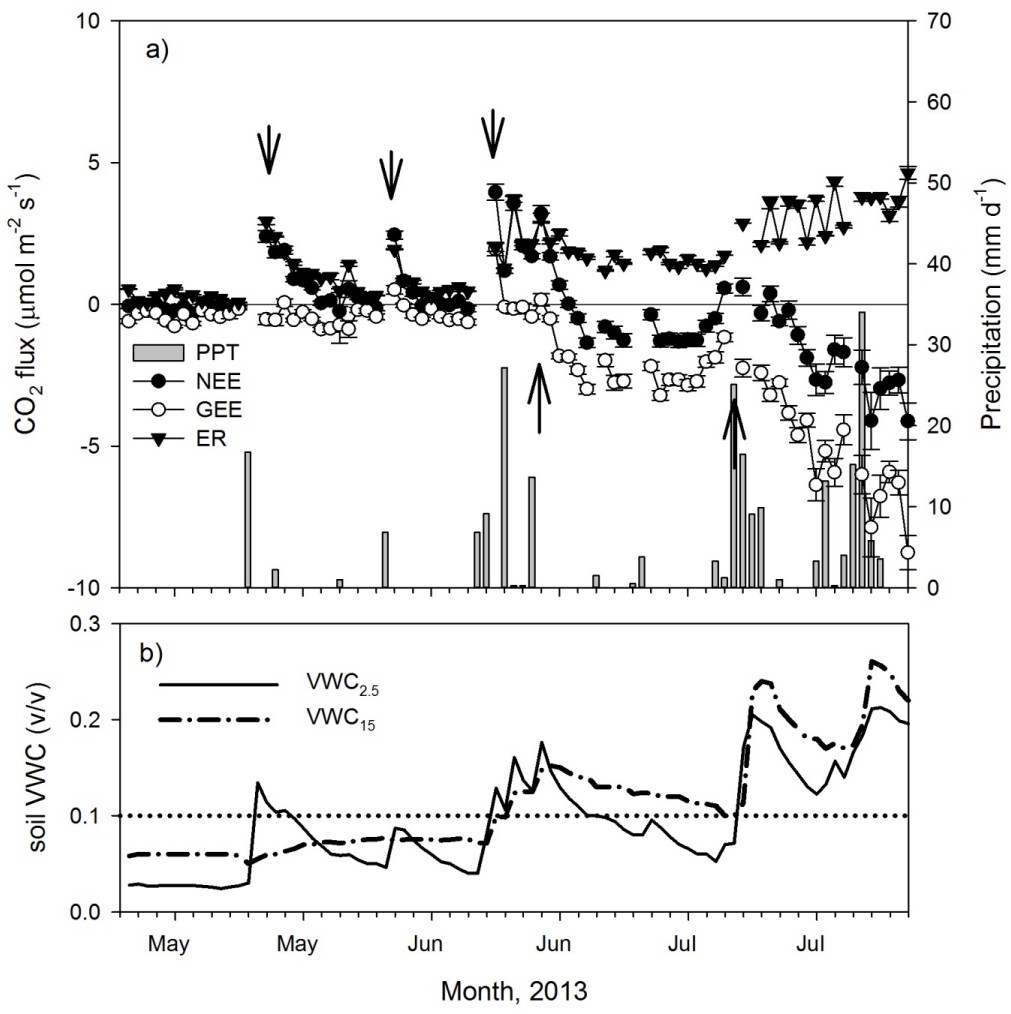


**Figure 3. Dynamics of a) precipitation (mm d$^{-1}$) and net ecosystem exchange (NEE, μmol m$^{-2}$ s$^{-1}$, daily means ± 1**


**SE) and its components, the gross ecosystem exchange (GEE, μmol m$^{-2}$ s$^{-1}$) and ecosystem respiration (ER, μmol m$^{-}$**


**$^{2}$ s$^{-1}$) for the transition from the dry (December – May) to the wet season (June – November) in 2013. b) volumetric**


**soil water content dynamics (VWC, v/v) at two depths (2.5 cm and 15 cm). Arrows indicate apparent changes in**


**GEE and ER trends. Dotted line indicates SWC = 0.1.**




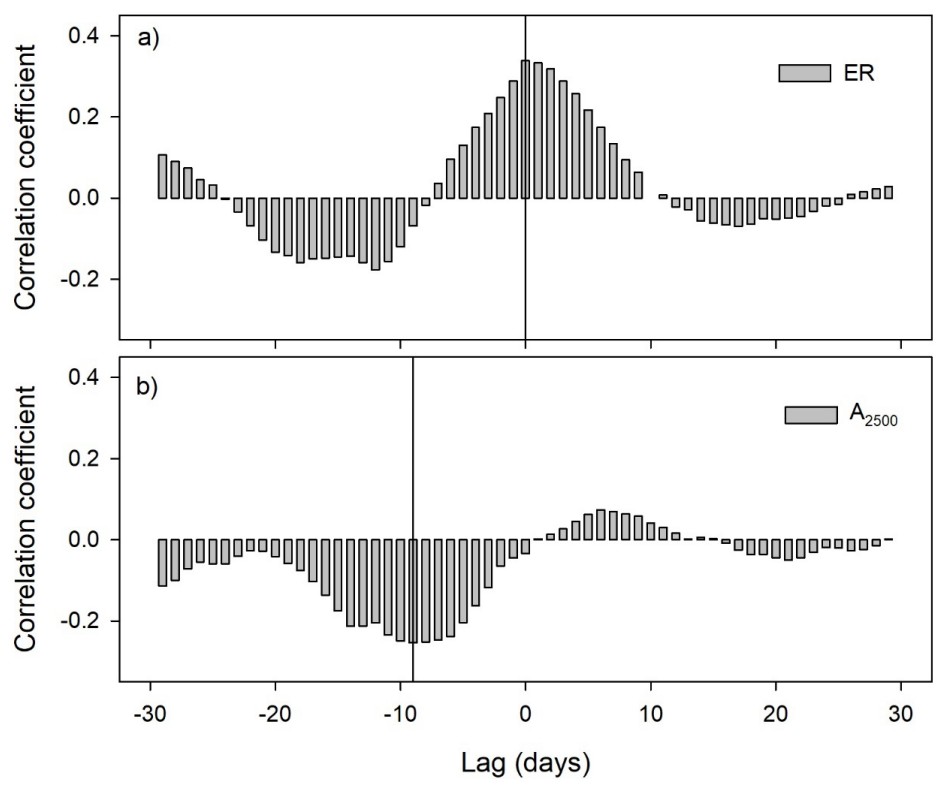


**Figure 4. Cross-correlation coefficients between detrended time series of soil water content at 2.5 cm depth and ecosystem respiration (ER, a), and between soil water content at 15 cm depth and photosynthesis at 2500 µmol m$^{-2}$ s$^{-1}$ of photosynthetic photon flux density (A$_{2500}$; b).**


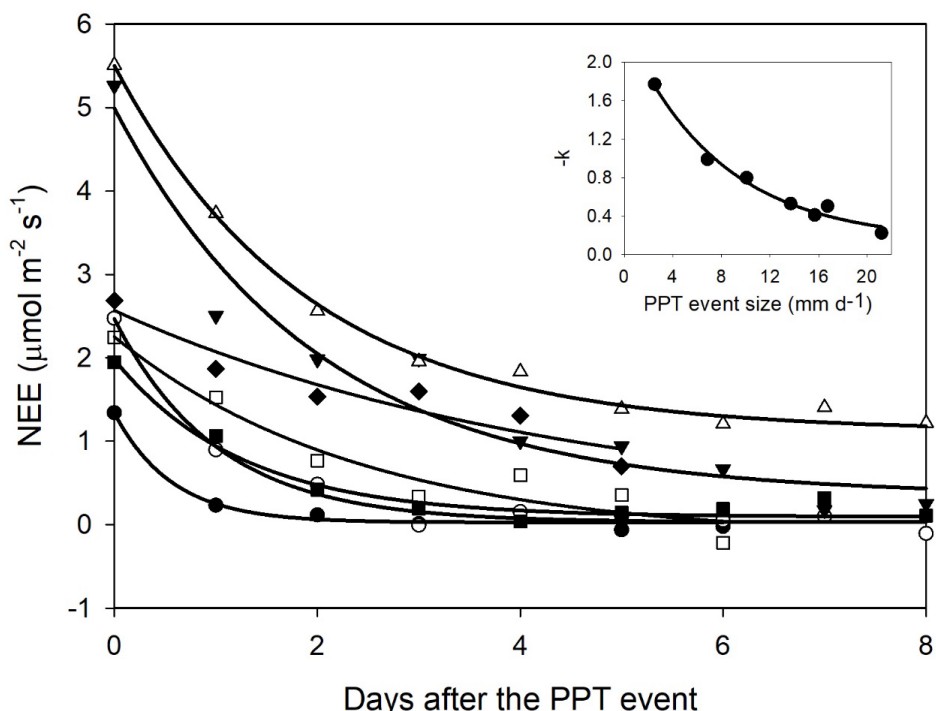


**Figure 5. Net ecosystem exchange (NEE) after a precipitation event showing the decreasing effect through time (days). The decreasing effect rate was adjusted to an exponential negative model NEE = yo + a \*exp(-k \*t). The insert stands for the relationship between the decaying rate (-k) and the PPT event that originated the NEE change. This relationship was fitted with an exponential model (black line; -k = yo + a \*exp(-b\*PPT_event). Symbols indicate different PPT event sizes that originated the NEE change, 13.7mm d[-1] (Δ), 16.74 mm d[-1] (▼), 6.86 mm d[-1] (○), 10.08 mm d[-1] (■),2.52 mm d[-1] (●), 21.18 mm d[-1] (□), and 15.68 mm d[-1] (♦).**






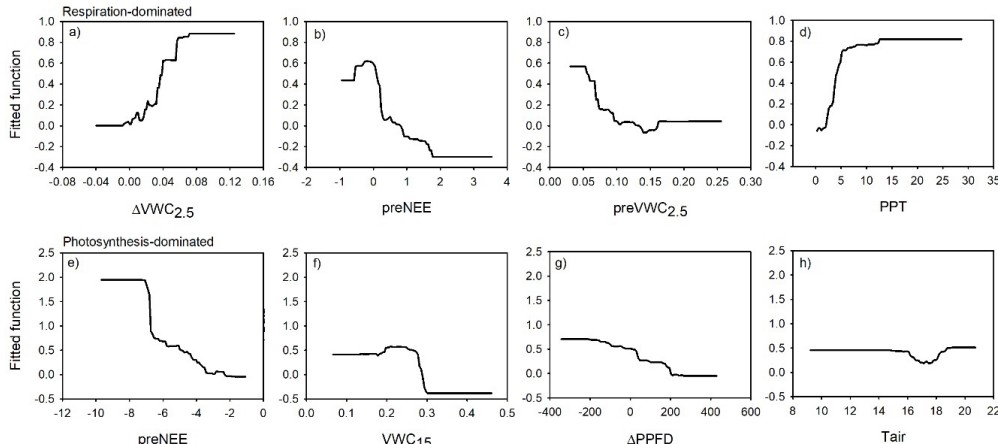


**Figure 6. Fitted functions of the boosted regression trees between the "priming CO2 effect" and the four most important environmental variables at ecosystem respiration-dominated period (upper panel) and at the photosynthesis-dominated period (bottom panel). Priming effect ($\Delta$NEE, $\mu$mol m$^{-2}$ s$^{-1}$); previous NEE (preNEE, $\mu$mol m$^{-2}$ s$^{-1}$); previous VWC at 2.5 cm depth (preVWC$_{2.5}$, v/v); PPT event size (PPT, mm); VWC at 15 cm depth (VWC15, v/v); change of photosynthetic photon flux density ($\Delta$PPFD, $\mu$mol m$^{-2}$ s$^{-1}$); air temperature (Tair, °C).**





**Appendix A. Ancillary figures**

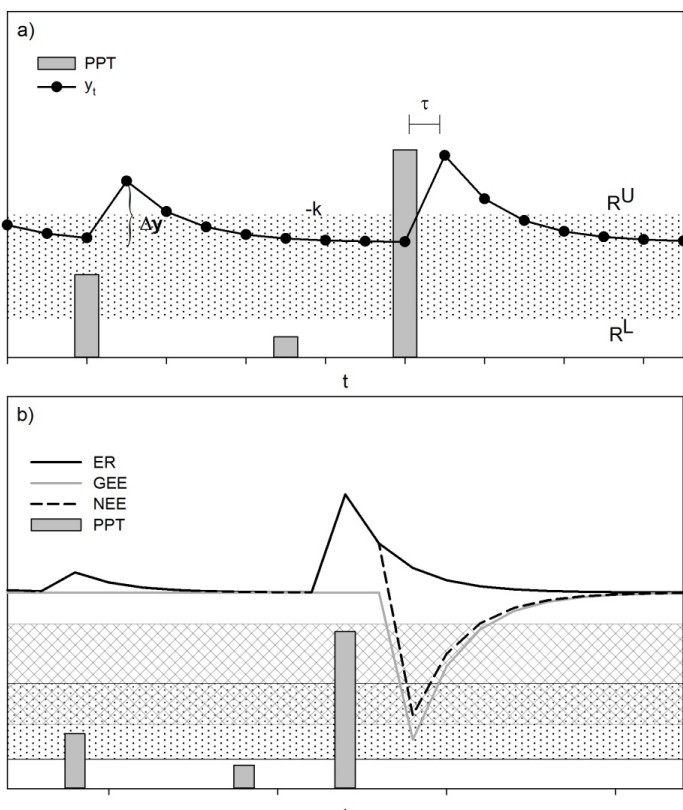


**Figure A1.** The Threshold-Delay model (Ogle and Reynolds, 2004). a) The magnitude of the increase in the
response variable ($\Delta t$, e.g., carbon flux, $y_t$) is determined by the size of PPT event and by the previous state of
the response variable. The decreasing rate of the response following the stimulus is denoted by –k. The low
PPT threshold ($R^L$) indicates the minimum size PPT event to stimulate a response, and the upper PPT
threshold ($R^U$) indicates PPT events that do not cause additional increment in the response variable. The time
interval between the stimulus and the response is described by $\tau$. b) The response of the net ecosystem
exchange (NEE), that is the balance between the gross ecosystem exchange (GEE) and ecosystem respiration
(ER), vary in response to changes of GEE and ER. According to the T-D model, GEE and ER have different
PPT thresholds (doted band and mesh stand for effective PPT events size for ER and GEE, respectively), with



ER responding to smaller size PPT events than GEE, therefore, small PPT events favor C release whereas
large PPT events stimulate net C uptake by the ecosystem. Differences of time responses between soil
microorganisms and plants to soil wet up led GEE and ER to differ in time delays (τ), with shorter time delays
for ER than GEE (Huxman et al., 2004a). The hypothetical curve for NEE and its components was calculated
introducing arbitrary parameters in the T-D model equations of Ogle and Reynolds (2004).





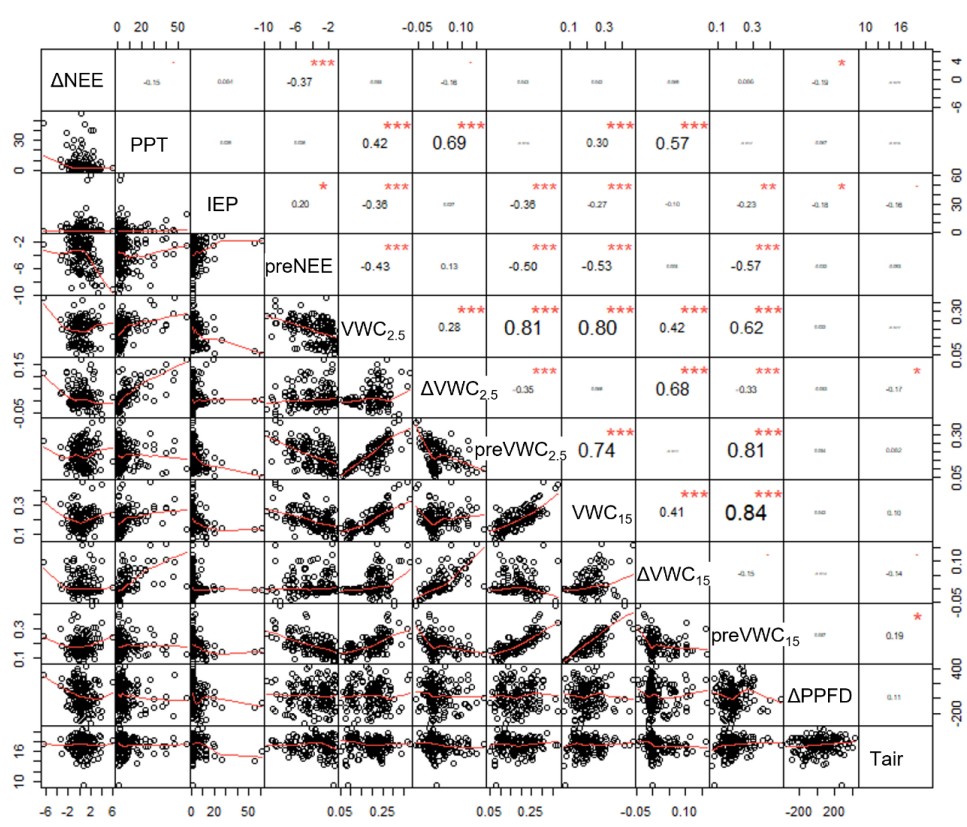


**Figure A2.** Correlation matrix among all variables.





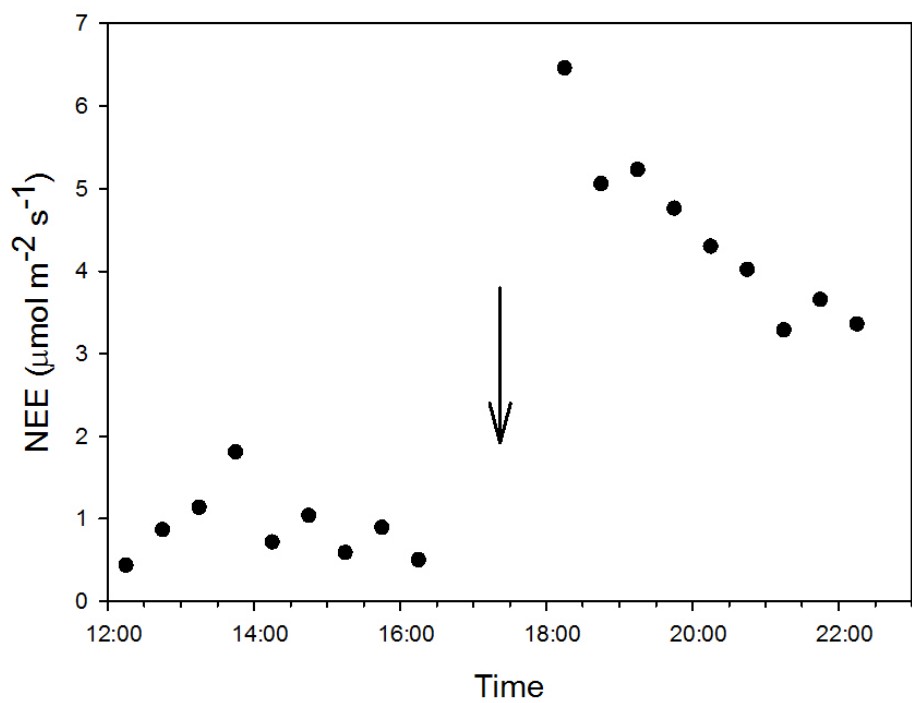


**Figure A3.** Dynamic of half an hour net ecosystem exchange ($\mu$mol m$^{-2}$ s$^{-1}$) after a precipitation event of 8.12

mm. The arrow indicates the time of PPT event occurrence.

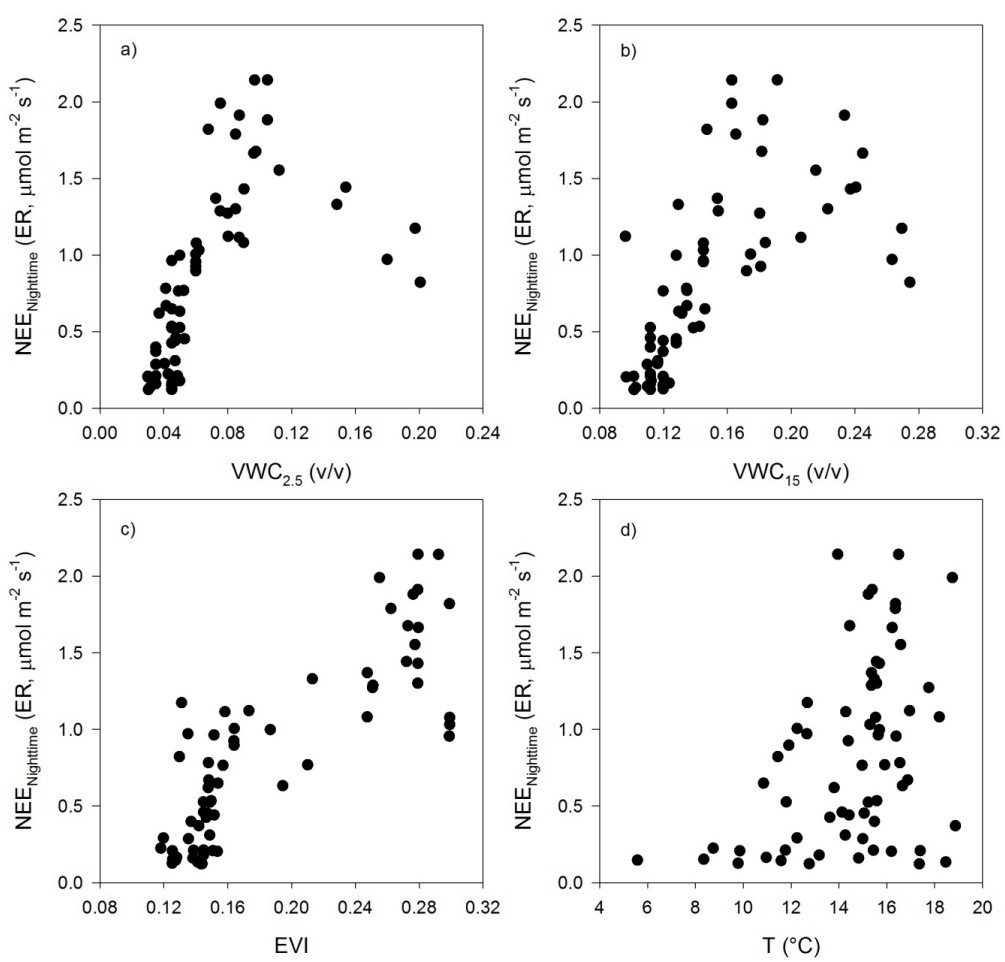


**Figure A4.** Relationship between nighttime-NEE derived ER and a) the soil volumetric water content at 2.5

cm depth ($VWC_{2.5}$, v/v), b) the soil volumetric water content at 15 cm depth ($VWC_{15}$, v/v), c) the enhanced

vegetation index (EVI), and d) the air temperature (T, °C).