# Peer review of "Dynamics of short-term ecosystem carbon fluxes induced by 1 precipitation events in a semiarid grassland. 2"

_Biogeosciences, 2022_

## Author Response (AR1)

Dear Dr. Paul C. Stoy
Associate Editor
*Biogeosciences*

We appreciate valuable comments and suggestions made by reviewers on our manuscript entitled: Dynamics of short-term ecosystem carbon fluxes induced by precipitation events in a semiarid grassland. We feel we have answered to all comments and recommendations. Please find below the response to each of their comments in *blue italics*.

We would like to express our great appreciation to you and the reviewers for the comments on our paper. If you have any further queries, please do not hesitate to contact us.

Best regards.

Josué Delgado Balbuena
CENID Agricultura Familiar

Anonymous Referee #1

Delgado-Balbuena investigate the response of the carbon cycle in a semiarid ecosystem in Mexico to precipitation using six years of eddy covariance data. They find that ecosystem respiration responds within hours in response to precipitation events as little as 0.25 mm but carbon uptake responds after a few days and required 40 mm or more of rain. This study aids in a conceptual understanding of the Birch effect via the threshold-decay model. The paper was well written an extremely interesting and publishable after considering only minor comments.

*Thank you for your valuable comments. Following, we addressed your suggestions.*

Minor comments:

107: "no rain" may be an overstatement, not sure.

*That it´s true in the worst cases. We adjusted this statement describing average values.*

*The region has a semiarid climate with mean annual precipitation of 424 mm ± 11 mm (last 30 years, Delgado-Balbuena et al., 2019) distributed mainly between June and September and with 6 – 9 moths of low PPT.  Winter-summer rain accounts for < 20% of the annual precipitation (Delgado-Balbuena et al., 2019).*

150: this is a somewhat large standard deviation threshold, but it probably makes sense to use it to not filter out small pulses.

*We apologize for this mistake. The threshold used for despiking was 6 standard deviations of a moving window (5 min), whereas any point beyond 8 SD was considered an outlier. The threshold for detecting "spikes" is some arbitrary; it goes from 3 – 8 (Burba, 2013) depending on site conditions. We considered to be less strict in the upper limit of standard deviation for detecting spikes whereas followed recommendations of flagging IRGA and CSAT malfunctions which generates majority of spikes; moreover, 30 min periods with gaps larger than 1% of total timeseries were eliminated for assurance of "high quality" fluxes. With a very strict despiking criteria, many "natural" fluctuations may be considered outliers, for instance, many of detected sonic temperature spikes may be subthermal structures into large-scale thermals.*

*The text was modified as follows:*

*Raw eddy covariance data were processed in EdiRe (v1.5.0.10, University of Edinburgh, Edinburgh UK).  Wind velocities, sonic temperature, $[CO_2]$, and $[H_2O]$ signals were despiked (Vickers and Mahrt, 1997), any value larger than six standard deviations into a moving*

*window (5 min) was considered a spike, whereas those values with a deviation larger than eight standard deviations were flagged as outliers.*

*Burba, G.: Eddy covariance method for scientific, industrial, agricultural, and regulatory applications: a field book on measuring ecosystem gas exchange and areal emission rates, LI-COR Biosciences, Lincoln, Nebraska, 331 pp., 2013.*

please use formal mathematical symbols in equation 2.

*We corrected the equation using symbols according to Lasslop et al. (2010), but conserved PPFD instead of Rg because we used the photosynthetic photon flux density (PPFD, $\mu mol$ $CO_2$ $m^{-2}$ $s^{-1}$) instead of global radiation. Also we conserved ER for ecosystem radiation instead of y to be consistent with ER through the manuscript.*

154: w' is defined here but used above in equation 1.

*This was specified in equation 1:*

*overbar denotes time averaging (30 min), and primes are the deviations of instantaneous values (at 10 Hz) of vertical windspeed (w', $ms^{-1}$) and molar volume of $CO_2$ ($CO_2$', $\mu mol$ $CO_2$ $m^{-3}$), from the block-averaged mean.*

160: 'advices of problem measurement due to rain events' should be rewritten.

*It was rewritten:*

*Fluxes were submitted to quality control procedures, i) stationarity (<50%), ii) integral turbulence characteristics (<50%), iii) flags of IRGA and sonic anemometer (AGC value<75, Max CSAT diagnostic flag = 63) which are frequently caused by raindrops on the anemometer transducers and IRGA path, iv) screening of flux values into a logical magnitudes (±20 $\mu mol$ CO2 m-2 s-1), and v) the u\* threshold was used to filter nighttime NEE under poorly developed turbulence. This threshold was defined through the 99% threshold criterion after Reichstein et al. (2005); it varied seasonally among years around 0.1 m $s^{-1}$.*

161: what was the basis for the 0.1 m/s ustar threshold? this is somewhat low for global ecosystems but representative for grasslands. The Reichstein algorithm assumes seasonal changes in its value (usually every 3 months), was this requirement softened here?

*We used the Reichstein algorithm for seasonal changes within each year. The ustar threshold was almost the same for all season-years, with an average of 0.1 $ms^{-1}$. Ustar values for each season/years were used to filter data.*

*We modified these lines to:*

*... and v) the u\* threshold was used to filter nighttime NEE under poorly developed turbulence.  This threshold was defined through the 99% threshold criterion after Reichstein et al. (2005); it varied seasonally among years around 0.1 m s$^{-1}$.*

equation 2 should also suffice to estimate changes in ER but its magnitude can often be better approximated using a non-rectangular hyperbola (but difficult to fit) or the Mitscherlisch model.

*We choose this model because of its simplicity. Since NEE was modeled at one-day window, a better fit was obtained by a simpler model. We tried for instance the modified hyperbolic model with ER depending on air/soil temperature (Gilmanov et al., 2007), but it failed more times than the single hyperbolic model to fit data.*

*Gilmanov, T. G., Soussana, J. F., Aires, L., Allard, V., Ammann, C., Balzarolo, M., Barcza, Z., Bernhofer, C., Campbell, C. L., Cernusca, A., Cescatti, A., Clifton-Brown, J., Dirks, B. O. M., Dore, S., Eugster, W., Fuhrer, J., Gimeno, C., Gruenwald, T., Haszpra, L., Hensen, A., Ibrom, A., Jacobs, A. F. G., Jones, M. B., Lanigan, G., Laurila, T., Lohila, A., G.Manca, Marcolla, B., Nagy, Z., Pilegaard, K., Pinter, K., Pio, C., Raschi, A., Rogiers, N., Sanz, M. J., Stefani, P., Sutton, M., Tuba, Z., Valentini, R., Williams, M. L., and Wohlfahrt, G.: Partitioning European grassland net ecosystem CO2 exchange into gross primary productivity and ecosystem respiration using light response function analysis, Agriculture, Ecosystems & Environment, 121, 93–120, https://doi.org/10.1016/j.agee.2006.12.008, 2007.*

Note minor usage errors like no period on line 282.

*All text was checked again for correcting this type of mistakes.*

Anonymous Referee #2

**General comments:**

This paper examines the dynamics of ecosystem respiration and productivity in response to precipitation events in a semiarid grassland in Mexico. The paper uses 6 years of eddy covariance measurements to investigate the effects of precipitation timing and magnitude on carbon fluxes. They found that ecosystem respiration has a rapid response to precipitation events and was triggered by events with as low as 0.25 mm of precipitation, while ecosystem carbon uptake only responded to events with >40mm of precipitation, and the response took several days. This paper is an interesting examination of the importance of the Birch effect to the overall ecosystem carbon balance in a semiarid ecosystem. As described below, there are some issues that should be addressed with minor revisions.

*We thank reviewer 2 for valuable comments. We addressed suggestions as follows:*

**Specific comments:**

There should be more details provided about the approach used for partitioning the flux data into GEE and ER. For example, when ER was estimated from the nighttime NEE data (line 166), was ER then extrapolated to daytime based on a temperature-dependent respiration model (as is commonly done), or is ER just taken as the average of the nighttime NEE?  In the results, it is not always clear which partitioning method was used for each analysis/figure. It would be good to include a comparison of the two approaches that were used. I also think it would be worth discussing how the partitioning approaches used here differ from commonly used partitioning approaches (e.g. Reichstein et al 2005, Lasslop et al 2010), and why these methods were chosen.

*This section was clarified:*

1) *Partitioning of GEE and ER form NEE was done through light-response curves. For this study we didn´t use the Reichstein et al. (2005) nor Lasslop et al., (2010) algorithms because we were interested in detecting changes at one day scale. Both algorithms use data windows larger than one day to estimate some parameters and tend to smooth fast changes in soil respiration like the observed in this study. We adjusted one light-response curve to NEE data per day.*
   *We added the next sentence:*

   *We choose this method instead of standard partitioning procedures (i.e. Reichstein et al., 2005 or Lasslop et al., 2010) because we were interested in detecting changes at one day scale. Both algorithms use data windows*

*larger than one day to estimate some parameters and tend to smooth fast changes in soil respiration like the observed in this study. For visually checking for changes in GEE and ER at diel time step, half-hours of NEE were partitioned by equation 2 and then averaged by day.*

2) *Mean ecosystem respiration (μmol m$^{-2}$ s$^{-1}$) derived from nighttime NEE was exclusively used for correlations with soil water content at two depths, air temperature and EVI (Fig. A4).*

3) *Along the manuscript we refer many times to ecosystem respiration when in fact data correspond to NEE, however, these are NEE fluxes during the dry season when there isn't plant activity, thus we considered these fluxes as exclusively ecosystem respiration from soil.*

4) *We specified in the statistical analysis that changes in NEE pre- and post-precipitation event were divided in "photosynthesis dominated" and "respiration dominated". This was done with the aim to separate larger contributions of photosynthesis and respiration without flux partitioning procedures that would smooth flux responses.*

*We modified the next paragraph of section 2.4:*

*Temporally integrated estimates are noted throughout this paper. Because GEE cannot be measured directly, it was estimated from light-response curves (see below), whereas ER was determined from i) light-response curves and ii) nighttime NEE data (under PPFD < 10 μmol m-2 s-1 light conditions). Henceforth, ecosystem respiration derived from light-response curves is denoted as "ER", and as "nighttime NEE" when derived from nighttime net ecosystem exchange data.*

More detail should also be given in section 2.5. Again, it is a bit unclear when daytime-derived vs nighttime-derived NEE data and ER were used. Also, it would be good to provide more detail about the distribution of gaps in the datasets. Since precipitation events generally cause quality issues in flux data, it would be helpful to quantify the distribution of data gaps around precipitation events (since these events are the focus of the study).

*Daytime NEE (without flux partitioning) was used for all analysis of changes in carbon fluxes induced by PPT events. When the dynamics of NEE, GEE and ER are described (Fig. 3a), we partitioned GEE and ER with the light-response curves, and then GEE, ER and NEE were averaged by day.*

1) *We added some lines in this section:*

*Mean ER derived from nighttime NEE data were used for analysis only when more than 50% of the data was available after QA/QC procedures. This data was exclusively used for correlation with environmental and soil data (see statistical analysis section). In contrast, daytime NEE (without partitioning) was used for the analysis of changes in NEE fluxes induced by PPT events.*

2) *As was described in methods (186-194), selection of precipitation events data was based on the well representativity of diel time pattern (>85% of NEE half-hours per day). This selection criteria included the pre and post event data. Thus, whether the previous day or the post-event day didn't have enough data, both days were flagged as bad data and discarded.*

*We added information in lines 264-267:*

*A total of 391 PPT events were isolated over the six years, but 34% did not accomplish with conditions of diel time representativity (>85% of NEE data); thus, 256 events were used for statistical analysis. A sample of 100 PPT events was used for the respiration dominated fluxes (>-1.0 µmol m-2 s-1) and 156 PPT events for the photosynthesis dominated fluxes (<-1.0 µmol m-2 s-1).*

Line 44: It would be good to include some references about this.

*References were added:*

*Consequently, the productivity and stability of these ecosystems are more vulnerable to changes in climate, particularly to changes of the historic mean annual PPT (MAP; Wang et al., 2021) amounts and the change in the periodicity (frequency) of these PPT events (Korell et al., 2021; Nielsen and Ball, 2015).*

*Korell, L., Auge, H., Chase, J. M., Harpole, W. S., and Knight, T. M.: Responses of plant diversity to precipitation change are strongest at local spatial scales and in drylands, Nat Commun, 12, 2489, https://doi.org/10.1038/s41467-021-22766-0, 2021.*

*Nielsen, U. N. and Ball, B. A.: Impacts of altered precipitation regimes on soil communities and biogeochemistry in arid and semi-arid ecosystems, Glob Change Biol, 21, 1407–1421, https://doi.org/10.1111/gcb.12789, 2015.*

*Wang, B., Chen, Y., Li, Y., Zhang, H., Yue, K., Wang, X., Ma, Y., Chen, J., Sun, M., Chen, Z., and Wu, Q.: Differential effects of altered precipitation regimes on soil carbon cycles in arid versus humid terrestrial ecosystems, Global Change Biology, 27, 6348–6362, https://doi.org/10.1111/gcb.15875, 2021.*

Lines 178-179: Were there unfilled gaps in the data used for the annual NEE estimates?

*As it's mentioned, data gaps shorter than two hours were linearly interpolated, whereas larger gaps were left as empty data. Half-hourly data pre and post PPT events was used only when more than 80% of daytime NEE was present (data after linear interpolation of <2 h gaps). These data were used to calculate mean daytime NEE.*

*In this study we did not report annual NEE, and when we compared C effluxes from priming effect to annual NEE, we cited to Delgado-Balbuena et al. (2019) where seasonal and annual C fluxes were estimated. For that study, NEE data were gapfilled with standard methods (Reichstein et al., 2005) for annual C balances. Moreover, when we used NEE partition like in Fig. 3 to check for dynamics of gross ecosystem exchange and ecosystem respiration, not gap-filled data was used for light-response curves fitting.*

*We added the next paragraph indicating how the contribution of the priming effect to annual carbon balances was estimated (L215-220):*

*To estimate the contribution of the priming effect to the annual carbon balance in the semiarid grassland, we averaged and extrapolated ΔNEE by the number of precipitation events per year. Decaying rates, PPT event size, and previous soil and flux conditions were not considered in this approach. Although this is a rough estimation, it provides a broad overview of how precipitation patterns influence the annual carbon balance. It is important to have this broad overview to better understand the impacts of climate change on carbon cycling in semiarid grasslands.*

Line 197: Citation for the MODIS data should be included.

*Citation was included:*

*Enhanced vegetation index (EVI) of 250 m spatial resolution and 8 day time-resolution from NASA's MODIS instruments (Didan, 2021) was used as an approximation of plant leaf activity. The Savitzky-Golay (Yang et al., 2014) filter was used to eliminate outliers of EVI derived from adverse atmospheric conditions.*

*Didan, K. MOD13Q1 MODIS/Terra Vegetation Indices 16-Day L3 Global 250m SIN Grid V061. NASA EOSDIS Land Processes DAAC. https://doi.org/10.5067/MODIS/MOD13Q1.061, 2021.*

**Technical corrections:**

There are a number of grammatical errors and typos throughout the paper, and some sentences that should be revised for clarity. I have noted some examples below, although this is not a complete list:

- Line 20: "PPT periodicity, magnitude" should be "PPT periodicity and magnitude"

*Done*

- Line 24: "GEE; such as" should be "GEE; where"

  *Done*

- Line 29: "significatively high respect to the carbon balance"– should this be "significantly high with respect to the carbon balance"?

  *Done*

- Line 31: change "by the climate change effect" to "due to the effects of climate change"

  *Done*

- Line 84: change "interested on" to "interested in"

  *Done*

- Line 191: In equation 3 it would be better to have "post-event" and "pre-event" as subscripts

  *Done*

- Line 179-182: This sentence should be clarified.

  *This sentence was rewritten:*

  *Data gaps shorter than two hours were linearly interpolated, whereas gaps larger than two hours were left as empty data. Only daytime-NEE data were used for most of the analysis because nighttime NEE is subjected to quality problems like poorly developed turbulence. Moreover, if mean NEE is estimated from only a few 30-minute nighttime NEE half-hours, the estimate may be biased if the full night cycle is not represented similarly across days.*

- Line 215: change "for identify" to "to identify"

  *Done*

- Line 274: change "enhanced" to "enhance"

*Done*

*All text was checked again for correcting this type of mistakes, and some sentences were modified to be clearer.*